CERN-TH-2021-197

# Probing multi-particle unitarity with the Landau equations

Miguel Correia[a,b], Amit Sever[c], and Alexander Zhiboedov[a]

[a]CERN, Theoretical Physics Department, CH-1211 Geneva 23, Switzerland
[b]Fields and Strings Laboratory, Institute of Physics,
École Polytechnique Fédérale de Lausanne, Switzerland and
[c]School of Physics and Astronomy, Tel Aviv University, Ramat Aviv 69978, Israel

We consider the $2 \to 2$ scattering amplitude of identical massive particles. We identify the Landau curves in the multi-particle region $16m^2 \leq s,t < 36m^2$. We systematically generate and select the relevant graphs and numerically solve the associated Landau equations for the leading singularity. We find an infinite sequence of Landau curves that accumulates at finite $s$ and $t$ on the physical sheet. We expect that such accumulations are generic for $s,t > 16m^2$. Our analysis sheds new light on the complicated analytic structure of nonperturbative relativistic scattering amplitudes.

*Dedicated to the memory of*
*Richard J. Eden and John C. Polkinghorne.*

## I. INTRODUCTION

Implementation of multi-particle unitarity is among the biggest challenges in the nonperturbative $S$-matrix bootstrap. This paper studies the "shadow" that multi-particle unitarity casts on the $2 \to 2$ amplitude.

It is a well-known fact that scattering amplitudes develop a nontrivial discontinuity along the normal thresholds. This fact is a direct consequence of unitarity. Once combined with analyticity, unitarity also predicts the existence of infinitely many curves in the $s-t$ plane along which the amplitude develops double discontinuity. These so-called *Landau curves* are more detailed characteristics of the amplitude's analytic structure. This paper explores the $2 \to 2$ scattering of identical scalar particles of mass $m$. The Landau curves found here should be present in any massive quantum field theory.

We assume that $m$ is the lightest particle in a theory. For simplicity we also assume that the theory has $\mathbb{Z}_2$-symmetry, such that the scattered particles are $\mathbb{Z}_2$-odd.[1]

We only concern ourselves with the behavior of the amplitude on *the physical sheet*. This is the region in the complex $s,t$ planes that is continuously connected to $0 < s,t,u < 4m^2$, without going through the multi-particle normal thresholds.

Let us quickly summarize the state-of-the-art results in this context, see figure 1. When one of the Mandelstam variables is in the elastic region, say $4m^2 \leq s \leq 16m^2$, unitarity relates the $2 \to 2$ amplitude to itself. Correspondingly, the Landau curves in this regime are known explicitly [1, 2], see appendix B.

On the other hand, in the regime where both $s,t > 16m^2$, unitary relates the discontinuities of the $2 \to 2$

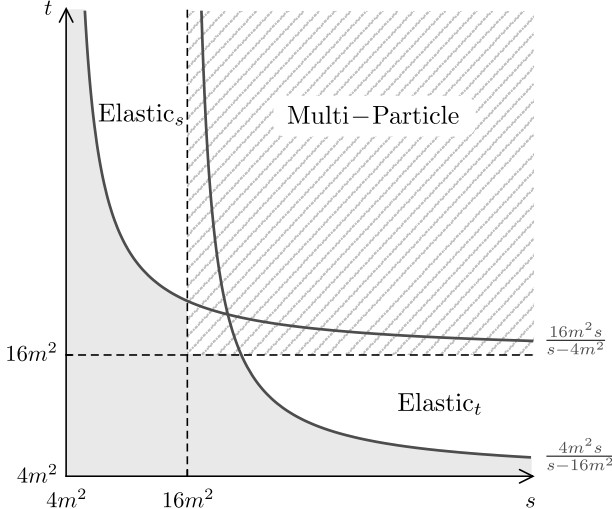

FIG. 1. The Landau curves in the elastic region $4m^2 \leq s,t < 16m^2$ are known thanks to elastic unitarity. The first of these are plotted in this figure. In the gray region, the double discontinuity is equal to zero. The main purpose of the present paper is to explore the structure of the Landau curves in the multi-particle region $s,t \geq 16m^2$.

amplitude to amplitudes with four particles or more. As a result, much less is known about the Landau curves in this multi-particle regime. In [3], five Landau curves in this regime were identified, out of which a few were found explicitly in [3, 4].

Using graph-theoretic tools implemented through a systematic computer search, we find all the Landau curves that asymptote to both, $t = 16m^2$ at large $s$ and to $s = 16m^2$ at large $t$.[2] Our results are summarized on figure 5 and figure 4. In particular, we find infinitely

_______

[1]For example, this applies to the pion scattering in QCD.

_______

[2]When claiming that the set of Landau singularities we find is complete, we will also assume that there are no bound states. By bound states we mean poles on the physical sheet in the region $0 < s,t,u < 4m^2$.

many Landau curves that accumulate towards the curve

$$(s - 16m^2)(t - 16m^2) - 192m^4 = 0 \,. \tag{1}$$

We expect such accumulation points to be a generic characteristic of multi-particle unitarity and that there are infinitely many of them at higher $s, t$, on the physical sheet.[3]

The plan of the paper is as follows. In section II we review the relation between analytically continued unitarity and the Landau equations. We also formulate the problem of finding the leading multi-particle Landau curves addressed in the present paper. In section III we present the solution to the problem. In section IV we collect implications of our results, future directions, and relation to other works. Many technical details are collected in the appendices.

## II. ANALYTICALLY CONTINUED UNITARITY AND THE LANDAU EQUATIONS

The $2 \to 2$ scattering process is characterized by an analytic function $T(s, t)$ that depends on two independent (complex) Mandelstam variables $s = -(p_1 + p_2)^2$ and $t = -(p_1 + p_4)^2$, where $p_i^\mu$ are the on-shell momenta, $p_i^2 = -m^2$, of the scattered scalar particles.[4]

We would like to understand the minimal set of singularities possessed by $T(s, t)$ as a consequence of unitarity and crossing. While the general answer to this question is beyond the scope of this paper, here we aim at revealing an infinite subset of singularities associated with multi-particle unitarity. The simplest singularities of this kind are normal thresholds. These are branch-point singularities at $s, t, u = (nm)^2$, with $n \geq 2$. Their presence follows directly from unitarity

$$\mathrm{Disc}_s T(s, t) \equiv \frac{T(s + i\epsilon, t) - T(s - i\epsilon, t)}{2i} = \sum_n \!\!\!\!\!\!\int_n T_{2 \to n} T^\dagger_{2 \to n} \,,$$

$$\text{with} \quad s \geq 4m^2, \quad 4m^2 - s < t < 0, \tag{2}$$

and the fact that $T_{2 \to n} = 0$ for $s < (nm)^2$. Here, the integral is over the $n$-particle phase space. To each term in the sum in (2) we can assign the graph in figure 2.a.

The vertices in this graph represent the amplitudes $T_{2 \to n}$, $T^\dagger_{n \to 2}$ and the lines between them represent the $n$-particle state.

As we analytically continue (2) to $t > 0$, we may encounter discontinuities of $\mathrm{Disc}_s T(s, t)$ in $t$. For example,

---

[3] This is in sharp contrast to the situation in the physical region where in every bounded portion of kinematic space only a finite number of singularities exists [5]. By *the physical region* we mean kinematics that can be directly probed in a scattering experiment.

[4] The results derived in this paper should equally apply to spinning particles.

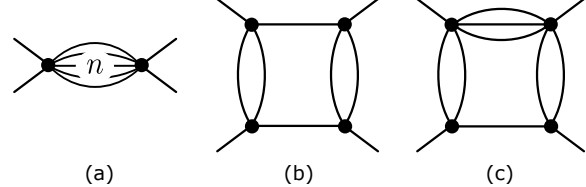

(a)        (b)        (c)

FIG. 2. A few simplest examples of graphs that represent various singularities of the $2 \to 2$ scattering amplitude. a) The bubble diagram represents multi-particle normal thresholds. b) The two-particle box diagram. It represents a Landau curve along which the scattering amplitude develops double discontinuity. c) The four-particle box diagram. This diagram corresponds to four-particle scattering both in the $s$- and in the $t$-channel. In this paper we systematically study the graphs of this type and the corresponding Landau curves.

consider the term in (2) with $n = 2$. Both $T_{2 \to 2}(s, t')$ and $T^\dagger_{2 \to 2}(s, t'')$ have a normal 2-particle threshold in the $t$-channel. These start to contribute to the corresponding phase space integral in (2) at a new branch-point that is located at

$$(s - 4m^2)(t - 16m^2) - 64m^4 = 0 \,, \tag{3}$$

along which the scattering amplitude develops double discontinuity, see [2] for details.

We can assign to this double discontinuity the graph in figure 2.b, where again, the lines represent (on-shell) particles and the vertices represent four-point amplitudes that have been analytically continued outside the regime of real scattering angles.

As we take $s > 16m^2$ more $n$'s contribute to (2) and more singularities are produced by the corresponding phase space integration. For example, the integration over the four-particle phase space ($n = 4$) can produce a cut of $\mathrm{Disc}_s T(s, t)$ in $t$ that results from the analytically continued two-particle normal threshold of $T_{2 \to 4}$ and $T^\dagger_{2 \to 4}$. The graph that represents this contribution to the double discontinuity $\mathrm{Disc}_t \mathrm{Disc}_s T(s, t)$ is plotted in figure 2.c.

Similarly, for any singularity that follows from multiple iteration of (analytically continued) unitarity we can associate a corresponding graph. By iteration of unitarity we mean the double discontinuity of the amplitude that is generated from a singularity of $T_{2 \to n}$ and another singularity of $T^\dagger_{2 \to n}$, through the analytic continuation of the phase space integration in (2) to $t > 0$. The singularities of $T_{2 \to n}$ and $T^\dagger_{2 \to n}$ themselves follows from analytically continued unitarity in a similar fashion. The graph that we associate to such a contribution to $\mathrm{Disc}_t \mathrm{Disc}_s T(s, t)$ is defined recursively, by gluing together a graph that represents a singularity of $T_{2 \to n}$ with a one that represents a singularity $T^\dagger_{2 \to n}$ with $n$-lines.

To enumerate all singularities that emerge in this way, we can go in the opposite direction and first enumerate all graphs that may result in a singularity of the am-

plitude. Whether a given graph leads to a singularity of the amplitude in a certain region in the complex $s, t$ planes is a kinematical question that does not depend on the details of the sub-amplitudes, represented by the vertices in the graph.[5] Hence, to answer this question we can equivalently take them to be constants. After doing so, it becomes evident that the same singularity, if it exists, is also generated by the Feynman diagram that coincides with the graph obtained from unitarity. The relevant singularity of the diagram comes from the region of loop integration where all propagators go on-shell [6, 7]. Other singularities of Feynman diagrams may result from a region of the loop integration where only a subset of propagators is on-shell. Those propagators that remain off-shell at the locus of a given singularity can thus be regarded as part of an higher point vertex that is not constant. For example, the Feynman diagrams that correspond to the graph in figure 2.a with two lines and the graph in figure 2.b, both have normal threshold at $s = 4m^2$. Hence, the set of all singularities of a Feynmann diagram includes the singularities of the corresponding graph and graphs obtained from it by collapsing some subset of lines into vertices with more legs. This operation is called a contraction.

If a generic diagram has an $n$-particle cut then it has a normal threshold starting at $n^2 m^2$ in $s$, $t$ or $u$ (depending on which external legs are considered incoming/outgoing). This can be seen by contracting the rest of the lines into a bubble diagram as in figure 2.a, with $n$ legs.[6]

In this way we immediately conclude that figure 2.b has normal thresholds at $s = 4m^2$, $t = 16m^2$, and figure 2.c at $s = 16m^2$, $t = 16m^2$.

In conclusion, to enumerate the singularities that follow from unitarity we can equally enumerate the singularities of Feynman diagrams.

In this classification, the Feynman diagrams are only used as a tool to study the location of kinematical singularities of a non-perturbative amplitude. For more than two intermediate particles, we find this tool more practical than directly analyzing the analytic continuation of the unitarity relation (2).

The locations of singularities of Feynmann diagrams can be found using the Landau equations. These are summarized in appendix C and we refer the reader, for example, to [9, 10] for a detailed review.

A *leading singularity* of a Feynmann diagram is a singularity that coincides with that of the corresponding graph.

Therefore we may restrict our dissection to singularities of this type only. The Landau equations that correspond to such a singularity are

1. All propagators are on-shell, $k_i^2 = m_i^2$, where the index $i = 1, \ldots, P$ labels all the propagators and $k_i$'s are oriented momenta that flow through them.

2. At each vertex $v$, the momentum is conserved, $\sum_{j \in v} \pm k_i^\mu = 0$, with $+ \ (-)$ for ingoing (outgoing) momenta.

3. For any loop $l$, the momenta satisfy
   $\sum_{j \in l} \pm \alpha_j k_i^\mu = 0$, with $+ \ (-)$ sign for momenta along (opposite) the orientation of the loop, and non-zero coefficients, $\alpha_i \neq 0$.

Two solutions that are related by an overall rescaling of the coefficients corresponds to the same singularity. We may therefore normalize them such that $\sum \alpha_i = 1$.

For any solution to these equations we can associate a story in complexified spacetime. In this story the Feynman parameters, $\alpha_i$, are the proper times of on-shell particles, $k_i^2 = m_i^2$, that propagate along the spacetime interval $\Delta x_i^\mu = \alpha_i k_i^\mu$. Every vertex represents a scattering of these particles that takes place at a point. The spacetime interval between two vertices should not depend on the path between the vertices. This means that for a closed path (i.e. a loop) we have $\sum_{i \in l} \Delta x_i^\mu = 0$.

No general answer is known to the question of which parts of the Landau curve lead to singularities on the physical sheet (which is our main interest here).

With present understanding, answering it requires a careful case-by-case analysis. There is however a special class of solutions to the Landau equations, called $\alpha$-*positive*, for which the singularity on the physical sheet is sometimes easier to establish.[7] These are solutions for which $\alpha_i > 0$.[8] Below we concern ourselves with $\alpha$-positive solutions only.

The double discontinuity $\text{Disc}_t \text{Disc}_s T(s, t)$ does not depend on the order in which the two discontinuities are taken. For example, the graph in figure 2.b can equally be interpreted as a contribution to the double discontinuity that comes about by first considering the four particle

---

[5]Note that the described way of generating new singularities from old ones involves analytic continuation of the amplitudes. It might happen that due to some special properties of the amplitude, the expected singularity is not there. Here we assume that this does not happens and expect the singularities which follow from unitarity to be generically present.

[6]In fact, the set of contractions only leads to a pair of single vertices if each side remains *connected* after the cut. In graph-theoretic terms this requires the cut to be *minimal* [8]. Physically, this is consistent with the fact that on the RHS of unitarity (2) only *connected* S-matrix elements participate.

[7]More precisely, this has only been shown for planar Feynman diagrams [11]. We believe that all $\alpha$-positive solutions found in this paper correspond to singularities on the physical sheet. However, we do not prove this for the non-planar graphs.

[8]Let us emphasize an important subtlety. In the literature the notion of $\alpha$-positive graphs typically involves an extra assumption: all $k_i^\mu$ are real, and $k_i^0 > 0$. These are the solutions of the Landau equations that capture singularities of the amplitude in *the physical region* [12]. On the other hand, the $\alpha$-positive graphs considered here are relevant for singularities of the $2 \to 2$ amplitude on *the physical sheet*.

contribution to $t$-channel unitarity and then plugging in the single-particle pole of the analytically continuation of $T_{2\to4}$ and $T_{2\to4}^\dagger$ in the $s$-channel. We can therefore group the Landau curves into families that are characterized by two integers $(n_s, n_t)$, which are the maximal number of particles in the $s$-channel and $t$-channel unitarity they can be obtained from.

In this paper we focus on the $(4_s, 4_t)$ family of double discontinuities. These are the ones that originate from the analytic continuation of unitarity (2) up until $n = 4$ in both channels. Physically, this corresponds to restricting energies to $s, t < 36m^2$.

We expect that all Landau curves in families with $(n_s \geq 4, n_t > 4)$ and $(n_s > 4, n_t \geq 4)$, which are not already included in the $(4_s, 4_t)$ family, to lay above the $(4_s, 4_t)$ family in the $s - t$ plane of figure 1.

## III. GRAPH SELECTION

We now describe our systematic method of finding the $\alpha$-positive Landau curves in the $(4_s, 4_t)$. A characteristic feature of a graph associated with such a curve is that any of its internal lines can be taken to be one of the four (or less) particles in the unitarity relation in either the $s$-channel or the $t$-channel. In other words, any leg of the graph should have a 2- or 4-particle cut in at least one of the channels.

The number of potentially contributing graphs is infinite. We study finitely many graphs with a fixed number of vertices, $V$, of fixed maximum vertex-degree $D$, and increase $V$ gradually. As we do so, the number of graphs to be analyzed grows factorially and the problem rather quickly becomes intractable.[9] To overcome this difficulty, we rule out graphs that a priori cannot possibly have $\alpha$-positive Landau curve or involve more than four particles in the $s$- or $t$- channel. Importantly, this selection process has a precise graph-theoretic implementation, so that it can be imposed before solving the Landau equations. Eventually we find that the number of the relevant graphs stabilizes at a handful number of graphs.

Throughout our analysis we assume $\mathbb{Z}_2$ symmetry of the amplitude that restricts the vertex degree $D$ to be even.

The set of criteria that we use to select the relevant graphs are as follows:

- We look for Landau curves in the $(4_s, 4_t)$ family. Correspondingly, we demand that any leg of a graph should have a four-particle or two-particle cut in at least one of the channels. Even though this criteria sounds very intuitive, we have not proved it. Instead, we will see that all the curves that result

from graphs that satisfy it pass through the region $16m^2 < s, t < 36m^2$ and asymptote to $16m^2$.[10]

- A graph only admits an $\alpha$-positive solution to the Landau equations if each of its subgraphs admits an $\alpha$-positive solution to the Landau equations.

  According to this criterion, we can discard a graph by identifying that one of its subgraphs cannot have an $\alpha$-positive solution.

- We can discard a graph if a subgraph of it can be contracted without affecting the solution. That is because the corresponding Landau curve if it exists, is already accounted for by the contracted graph.

We denote *trivial sub-graph* a graph that falls into one of the last two categories. We have identified a few families of trivial graphs that involve bubbles, triangles, and boxes.[11] They are discussed in appendix D.

Computationally, we found it most efficient to proceed as follows

1. We start by generating all graphs without trivial bubbles, with fixed number of vertices $V$ that contain at least one vertex of degree $D$, but no vertices of higher degree.

2. We discarded graphs with trivial sub-triangles.

3. We discarded graphs without 2-particle or 4-particle cuts in at least two channels.

4. We selected the graphs for which all legs can be cut by 2-particle or 4-particle cuts.

5. We discard graphs with trivial sub-boxes.

6. Finally, we select graphs for which an $\alpha$-positive solution to the Landau equations is found.

Step 1 was implemented using *nauty and Traces* [13]. Steps 2-5 were implemented using the open source network analysis package *igraph* [14], adapted to `mathematica` by the *igraph/M* package [15]. Step 6 was implemented in `mathematica`. The details of the implementations of each step can be found in appendix E.

As we increase $D$, it is harder to satisfy the four-particle constraint and the absence of trivial sub-graphs. In particular, we did not find any graphs satisfying our criteria with $D \geq 8$. In table I, we list the number of graphs at each step for $D = 4$ and $D = 6$.

As can be seen from table I, considerable reduction in the number of graphs occurs at step 4 as the number of vertices increases. At this order it becomes an incredibly

---

[9]For $D = 4$ we analyzed all the graphs with $V \leq 12$ and for $D = 6, 8$ up to $V \leq 8$.

[10]Conversely, no graph that was left out by this criterion was found to have a curve in this region. This check was made until $V = 8$ for $D = 4$ and until $V = 6$ for $D = 6$.

[11]These are subgraphs with 2, 3, and 4 vertices correspondingly.

| Quartic graphs ($D = 4$) | | | | | | | | |
|---|---|---|---|---|---|---|---|---|
| # of vertices ($V$) | 4 | 5 | 6 | 7 | 8 | 9 | 10 | 11 | 12 |
| All graphs (no trivial bubbles) | 2 | 3 | 23 | 111 | 788 | 5639 | 46603 | 410114 | 3587793 |
| No trivial triangles | 2 | 1 | 10 | 33 | 232 | 1522 | 12696 | 113034 | 1023415 |
| 2- or 4-particle cuts (exist) | 2 | 1 | 7 | 25 | 157 | 955 | 7070 | 54835 | 429093 |
| 2- or 4-particle cuts (all legs) | 2 | 1 | 4 | 5 | 12 | 7 | 10 | 7 | 9 |
| No trivial boxes | 2 | 1 | 3 | 4 | 9 | 4 | 4 | 3 | 3 |
| $\alpha$-positive Landau curves | 2 | 1 | 2 | 1 | 3 | 1 | 1 | 1 | 1 |

| Quartic & sextic graphs ($D = 6$) w/ sextic vertex | | | | | |
|---|---|---|---|---|---|
| # of vertices ($V$) | 4 | 5 | 6 | 7 | 8 |
| All graphs (no trivial bubbles) | 9 | 109 | 2678 | 73918 | 2477395 |
| No trivial triangles | 6 | 22 | 553 | 14714 | 538309 |
| 2- or 4-particle cuts (exist) | 1 | 3 | 27 | 476 | 10356 |
| 2- or 4-particle cuts (all legs) | 1 | 0 | 2 | 1 | 3 |
| No trivial boxes | 1 | 0 | 1 | 0 | 0 |
| $\alpha$-positive Landau curves | 1 | 0 | 1 | 0 | 0 |

TABLE I. The number of graphs with vertex degree 4 (top) and vertex degree $\leq 6$ with at least one sextic vertex (bottom). Each column specifies the number of vertices and each row specifies a reduction step. We show the total number of $2 \to 2$ graphs without trivial bubbles in the second row. In the third row, graphs with trivial triangle subgraphs have been discarded. Next, we require that all legs can be put on-shell with at most 4-particle cuts since we are looking for Landau curves in the $(4_s, 4_t)$ family. We first demand that there exists at least one such cut of the diagram in each channel (fourth row). Next we demand that every line can be cut with a 2-particle or a 4-particle cut (fifth row). As a next step we discard the graphs containing a trivial box subgraph (sixth row). The last row has the number of graphs for which an $\alpha$-positive solution has been found by numerically solving the Landau equations. For quartic graphs, this number does not go to zero as the number of vertices is increased. This is due to an infinite family of diagrams generated by consecutive insertion of triangles (see figure 4). The corresponding Landau curves accumulate at finite $s, t$, see figure 5. Current computational limitation prevents us from increasing the number of vertices any further, but we believe that all the relevant diagrams have been identified.

tight criterion, but also very computationally expensive. For comparison, given the same set of graphs, we observe step 3 to be roughly a hundred times faster and step 2 around a thousand times faster.

Step 3 is logically included in step 4. Even though it is not necessary, it reduces total computing time.

Interestingly, step 2, the elimination of trivial sub-triangles, is essential to observe the quench in the growth of diagrams. We observe that the number of diagrams with trivial triangles that survives the criterion imposed by step 4 grows at least exponentially with the number of vertices.

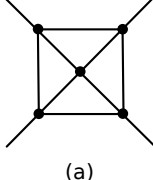 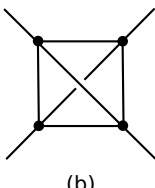

(a)      (b)

FIG. 3. The planar cross and the non-planar cross (open envelope) graphs. Each of the diagrams is the first one in an infinite chain of diagrams, see figure 4, that generates the Landau curves on the physical sheet, in the region $16m^2 \leq s, t < 36m^2$.

Let us now summarize our findings. Figures 4 and 5 depict all the graphs that satisfy $\alpha$-positive Landau equations with at most four particles in the $s$- and $t$-

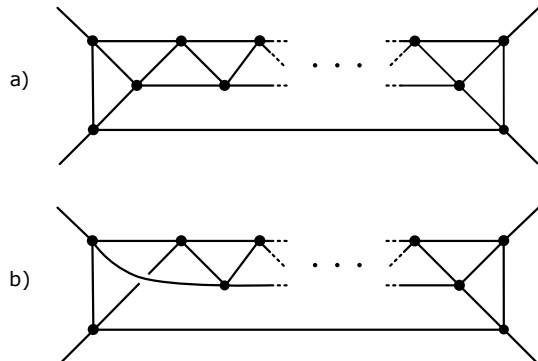

FIG. 4. The planar, a), and non-planar, b), triangle chain graphs. Remarkably, each of the graphs involves four-particle scattering both in the $s$- and in the $t$- channel. As the number of triangles grows, the corresponding Landau curves quickly accumulate around the locus (1) on the physical sheet. Notice that adding a single triangle to each chain increases the number of vertices $V$ by 2. Closely related diagrams appeared before in [11, 16, 17].

channels, and figure 5 the corresponding Landau curves. The graphs in figure 4 and figure 5, as well as the Landau curves in figure 5 are the main results of the paper.[12]

―――――

[12] All the curves that we found cross the region $16m^2 < s, t < 36m^2$.

In appendix A we exhibit the equations for some of the multi-particle Landau curves depicted in figure 5.

Interestingly, we find Landau curves crossing $16m^2 \leq s, t < 36m^2$ which originate from graphs with arbitrarily large $V$. These are depicted in figure 4 and their Landau curves (shown in black on figure 5) accumulate to the red curve on figure 5. This is a new feature compared to the elastic region $4m^2 \leq s, t < 16m^2$, where every bounded region of the kinematic space contains a finite number of Landau curves. We believe that this feature is characteristic for the multi-particle region and there are infinitely many accumulation points of the Landau curves there. We discuss this further below.

## IV. DISCUSSION

In this paper we have analyzed analytic properties of the $2 \rightarrow 2$ scattering amplitude on the physical sheet. In particular, we have focused on the leading Landau curves in the multi-particle region which originate from the analytic continuation of four-particle unitarity both in the $s$- and $t$- channels (in our notations $(4_s, 4_t)$ curves). Here we discuss various implications of our results as well as some interesting directions to explore.

### A. Lighest particle maximal analyticity

Eventually we would like to fully understand the analytic properties of the scattering amplitude on the physical sheet. In the context of perturbation theory a rich structure of singularities has been discovered already in a $2 \rightarrow 2$ scattering amplitude. These include anomalous thresholds [18–20][13], crunodes, acnodes and cusps [23]. No systematic understanding of these latter singularities exists up to this day.

Nevertheless in the course of these explorations a remarkable hypothesis has emerged. It concerns the $2 \rightarrow 2$ scattering of the lightest particles in a gapped theory (which is the subject of the present paper) and can be stated as follows.

**Lighest Particle Maximal Analyticity:** The $2 \rightarrow 2$ scattering amplitude of the lightest particles in the theory, $T(s,t)$, is analytic on the physical sheet for arbitrary complex $s$ and $t$, except for potential bound-state poles,

a cut along the real axis starting at $s = 4m^2$, and the images of these singularities under the crossing symmetry transformations.

Establishing this hypothesis even within the framework of perturbation theory is an important, open problem in $S$-matrix theory. Assuming lighest particle maximal analyticity (LPMA), the analytic structure of the $2 \rightarrow 2$ amplitude is concisely encapsulated by the Mandelstam representation.[14] From the point of view of our analysis, the nontrivial fact about LPMA is that scattering of lightest particles contains infinitely many subgraphs that by themselves do not respect maximal analyticity. For example, some of the trivial boxes subgraphs depicted in figure 19 do not admit the Mandelstam representation [24]. For LPMA to hold, embedding these subgraphs inside a larger graph that describes scattering of the lightest particles in the theory should render the complicated singularities of the subgraph harmless on the physical sheet. We have not studied the mechanism of how this happens, and we leave this important question for future work.

LPMA is a working assumption in some of the recent explorations of the S-matrix bootstrap, see e.g. [2, 25–27]. It is also one of the main reasons we have restricted our study to the physical sheet.

It would be very interesting to revisit the problem of establishing LPMA in perturbation theory. For example, showing that all the graphs considered in the present paper admit Mandelstam representation might provide a clue as to why it is valid more generally.

### B. Analytic continuation of multi-particle unitarity

As we discussed at the beginning of the paper, a direct way to see the emergence of double discontinuity of the amplitude is to analytically continue the unitarity relations (2) which involves $T_{2 \rightarrow n}$ scattering amplitude. While for $n = 2$ this has been done already by Mandelstam [1], very little work has been done for $n > 2$. Let us mention that some progress has been made for $n = 3$ in the papers [28, 29] but the connection between analytically continued multi-particle unitarity and the multi-particle Landau curves has not been explored systematically. It could be useful, for example, to better understand the lower bound on particle production along the lines of [2].

### C. Lightest particle $\alpha$-positive Landau curves

All the Landau curves discussed in the present paper satisfy the following properties:

---

We believe that the presented here list of curves in this region is complete. Showing this requires proving some further properties of the $\alpha$-positive Landau curves which we discuss in the conclusions. It also requires making sure that non $\alpha$-positive solutions to the Landau equations do not lead to the singularities on the physical sheet.

[13] Anomalous thresholds do not arise in the $2 \rightarrow 2$ scattering of the lightest particles, see e.g. [11] for a perturbative argument. However they are present in the $3 \rightarrow 3$ (or $2 \rightarrow 3$) scattering, see [21, 22].

---

[14] The Mandelstam representation involves an extra assumption that the discontinuity of the amplitude is polynomially bounded on the physical sheet.

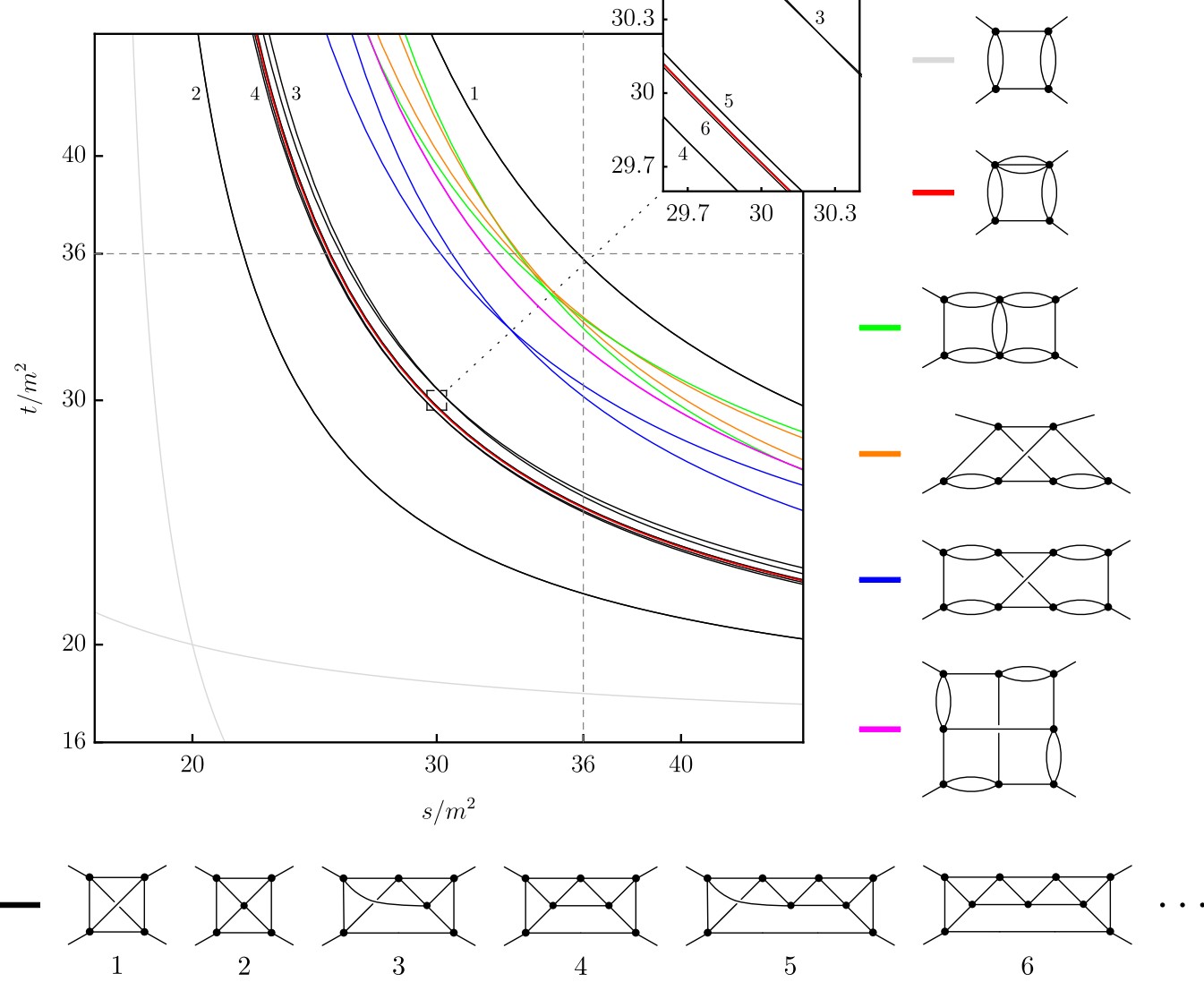

FIG. 5. The Landau curves in the $2 \to 4$ multi-particle region. To each diagram corresponds a pair of crossing symmetry-related curves. For crossing symmetric diagrams, there is only one curve. The red curve, given by equation (1), is an accumulation point of infinitely many Landau curves. The uppermost curve (black #1) is given by the non-planar cross diagram, figure 3 (b), while the planar cross, figure 3 (a), gives the lowermost curve (black #2). The planar triangle chain graphs (#2, #4, #6,...), figure 4 (a), approach the red curve from below while non-planar triangle chain graphs (#1, #3, #5,...), figure 4 (b), approach it from above. As shown in the inset panel, the approach is fast (see table II). We expect that the Landau curves presented here are all the curves that cross the square $16m^2 < s, t < 36m^2$ on the physical sheet. We collect explicit equations for some of the Landau curves in appendix A.

| Triangle chain curves at symmetric point: $s = t = x_n$. | | | | | | | | | | |
|---|---|---|---|---|---|---|---|---|---|---|
| $n$ | 1 | 2 | 3 | 4 | 5 | 6 | 7 | 8 | $\cdots$ | $\infty$ |
| $x_n$ | 35.8885 | 27 | 30.2385 | 29.7511 | 29.8799 | 29.8504 | 29.8579 | 29.8560 | $\cdots$ | 29.8564 |
| $\left\lvert \frac{x_n - x_\infty}{x_{n-1} - x_\infty} \right\rvert$ | - | 0.4735 | 0.1338 | 0.2756 | 0.2235 | 0.2533 | 0.2482 | 0.2547 | $\cdots$ | ? |

TABLE II. Accumulation of Landau curves at finite $s$ and $t$. The table lists the symmetric point ($s = t$) of the first few Landau curves produced by the infinite set of triangle chain diagrams (depicted in black in figure 5). The curves accumulate towards the red curve in figure 5. The last row of the table indicates that the approach towards the limiting curve is very quick (approximately geometric).

1. Asymptotic to normal thresholds. As $t$ (or $s$) goes to infinity, $s$ (or $t$) approaches normal thresholds.

2. Monotonic. For $s > 4m^2$ we have $\frac{dt}{ds} < 0$.

It is tempting to conjecture that in the context of *the lightest particle scattering* the properties above fully capture the nonperturbative analytic structure of the $2 \to 2$ amplitude on the physical sheet. Assuming it is true, the Landau curves form a simple hierarchical structure, where the $\alpha$-positive curve (if it exists) of a graph with a minimum cut across $n_s$ legs in the $s$-channel, and a minimum cut across $n_t$ legs in the $t$-channel has support in the region $s > (n_s m)^2$, $t > (n_t m)^2$. It then follows that the curves found in the present paper are complete in the region $16m^2 \le s, t < 36m^2$. This is an extra assumption to which we referred in the footnote 12.

### D. Extended elastic unitarity region

The results of this paper strengthen the picture in which the double discontinuity vanishes below the first elastic Landau curves

$$\rho(s,t) \equiv \mathrm{Disc}_t \mathrm{Disc}_s T(s,t) = 0 \tag{4}$$

$$\text{for} \quad s,t \ge 4m^2 \quad \text{and} \quad \begin{array}{l} (s-4m^2)(t-16m^2) < 64m^4 \\ (t-4m^2)(s-16m^2) < 64m^4 \end{array}.$$

We see that the multi-particle Landau curves do not spoil this relation and, therefore, we expect it to hold non-perturbatively.

Another outcome of our analysis is an extended region of validity of the analytically continued elastic unitarity. Let us consider the first multi-particle Landau curve that we encounter as we enter the region $s, t > 16m^2$. It is the planar cross curve given by the equation [3] (black curve #2 in figure 5)

$$s^3(t-16) + t^3(s-16) + 24\,s\,t(s+t-18) - 2s^2 t^2 = 0. \tag{5}$$

Let us call $\{4m^2 < s, t < \text{planar cross}\}$ *the extended elastic unitarity region*. In this region the double discontinuity $\rho(s,t)$ satisfies the following relation

Extended elastic unitarity :

$$\rho(s,t) = \rho^{\mathrm{el}}(s,t) + \rho^{\mathrm{el}}(t,s), \quad 4m^2 < s, t < \text{planar cross}, \tag{6}$$

where $\rho^{\mathrm{el}}(s,t)$ is the double discontinuity given by the Mandelstam equation in the $s$-channel, which expresses analytically continued elastic unitarity.

Notice that the equation (6), on one hand, involves only the $2 \to 2$ scattering amplitude. On the other hand, its origin lies in the details of multi-particle unitarity.

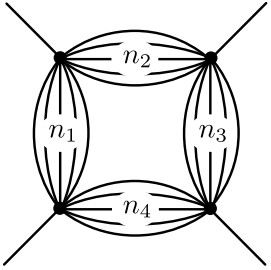

FIG. 6. A box graph with bridges of size $(n_1, n_2, n_3, n_4)$. The Landau curve that is associated to this graph is expected to be an accumulation curve of infinitely many Landau curves. They correspond to graphs that are obtained from this one by replacing an $n_i$-bridge with an $n_i$-chain.

### E. Accumulation points of the Landau curves are generic

We believe that the basic mechanism found in the present paper for the accumulation of the Landau curves on the physical sheet is generic. For example, we expect that the Landau curves that originate from the graphs depicted in figure 6 are accumulation points of infinitely many Landau curves on the physical sheet.

The basic mechanism is the one we observed for the triangle chains depicted in figure 4. By exchanging an $n_i$-particle bridge with a chain of $n_i$-particle sub-graphs, an infinite family of Landau curves is generated. As the length of the chain is increased, it is natural to expect that the solution of the Landau equations, if it exists, converges to the one of the $n_i$-particle bridge. In particular, we have already established existence of the $\alpha$-positive solution to the Landau equations for a chain of triangles. The triangle chain can now be exchanged with any $n_i = 3$ bridge of a diagram with an $\alpha$-positive solution to produce an accumulation sequence.[15]

This scenario leads to a very complicated structure of the Landau curves on the physical sheet. We will have an infinite number of accumulation points, accumulation points of accumulation points, etc.

### F. Higher multi-particle Landau curves

Our method is systematic and, given enough computational ability, can be used to find Landau curves above $s, t > 36m^2$. Graph selection should now involve $6-$particle (or heavier) cuts.

─────────

[15] Note that the triangle chain diagrams correspond to embedding $3 \to 3$ elastic scattering amplitude inside a $2 \to 2$ process. Similarly, the mechanism for accumulation of the Landau curves described above seems to be related to the behavior of the $n \to n$ multi-particle amplitudes close to the normal multi-particle thresholds. It would be interesting to study this behavior in more detail (for a related discussion see [17]).

It is likely, however, that the elimination of trivial bubbles and triangles will no longer be sufficient to quench the growth of the number of graphs. Unfortunately, the condition for larger subgraphs to be trivial is not so simple (see appendix D where we identify some trivial boxes). The problem becomes more complicated the deeper one delves into the multi-particle region. One may need to discard multiple-loop subgraphs, for example.

Direct use of unitarity may be a viable alternative, as shown in the elastic region. However, as mentioned earlier, analytic continuation of multi-particle kernels is a difficult task. The accumulation mechanism discussed here, which should follow from 4-particle unitarity, indeed does not indicate otherwise.

### G. $S$-matrix bootstrap applications

Our results have implications for the $S$-matrix bootstrap program. Indeed, the stumbling block of the current incarnation of the $S$-matrix bootstrap in $d > 2$ is that it was not possible so far to include the multi-particle amplitudes in the analysis.

Here we took an alternative route, where we tried to understand how the presence of multi-particle unitarity is reflected on the structure of the $2 \to 2$ amplitude. In some sense, we can think of the Landau curves found in the present paper as seeing multi-particle shadows on the elastic scattering wall.

Implementing the structure of the few leading Landau curves in the analytic structure of the amplitude will already be a step forward compared to some of the current explorations of the $S$-matrix bootstrap [25–27, 30]. Indeed, even (4) has not been realized in this context.[16]

Another interesting question is to what extent the detailed analytic structure is relevant for the low-energy observables, e.g. a few low-energy Wilson coefficients. We do not know the answer to this question, but recent works [25, 26] suggest that the low-energy observables are not very sensitive to that. It would be very interesting to better understand the origin of this phenomenon.

### H. Other future directions

A few other future directions are

- An interesting generalization of our analysis is to relax $\mathbb{Z}_2$ symmetry. Effectively it allows vertices of odd degree $D$ and will lead to new graphs and the corresponding Landau curves. It would be interesting to understand them in detail.

---

[16]The fixed point unitarity methods [31] do realize this structure but these have not been implemented in $d > 2$ yet [32].

- It would be interesting to apply techniques recently developed in [33] to better understand the approach of the Landau curves to the accumulation point, as well as the analysis of [34] to better understand the nature of multi-sheeted analytic structure of the corresponding Feynman graphs.

- It would be important to prove that all the curves we have found are indeed present on the physical sheet. For the planar graphs it is guaranteed, see [10, 11, 35]. For the non-planar cross in figure 3.b, the question was addressed in [36]. For the other non-planar graphs an extra analysis is required, either by following the analytic continuation of the corresponding Feymann integral or by directly analyzing the 4-particle unitarity kernel. It is also important to prove that non $\alpha$-positive solutions to the Landau equations do not lead to singularities on the physical sheet. Finally, one would like to show that there is a unique $\alpha$-positive solution associated with each nontrivial graph (which we assumed to be the case in appendix C).

- Our results should emerge from the flat space limit of the theory in AdS [37–41]. In the latter case, one computes the conformal correlators on the boundary of AdS. The double spectral density of the amplitude corresponds to the quadruple discontinuity of the corresponding correlator [42]. Complexity of the multi-particle scattering in this case translates into the complexity of the $n$-twist operators with $n > 2$, [43, 44].

- For large $N$ confining gauge theories, a new classical description emerges at large $s, t \gg m^2$ [45]. Can there also be a universal classical description in this regime for the non-perturbative amplitude? A crucial step in answering this question seems to be a better understanding of the analytic structure in this regime. Understanding this regime is necessary for establishing the Mandelstam representation of the scattering amplitude non-perturbatively. It is also important for developing possible truncation schemes in which the complicated multi-particle unitarity structure can be simplified.

### ACKNOWLEDGMENTS

We thank Aditya Hebbar, Kelian Häring, José Matos, Andrew McLeod, João Penedones, Slava Rychkov, and Kamran Vaziri for useful discussions. This project has received funding from the European Research Council (ERC) under the European Union's Horizon 2020 research and innovation programme (grant agreement number 949077). AS was supported by the Israel Science Foundation (grant number 1197/20).

## Appendix A: Multi-particle Landau curves

Here we collect explicit equations for some of the multi-particle Landau curves discussed in the paper. For convenience we set the mass $m = 1$. We refer to the various diagrams by their colors as depicted on figure 5.

**Red curve:** (the accumulation curve)

$$(s - 16)(t - 16) - 192 = 0. \tag{A1}$$

**Planar cross:** (see figure 3.a, and [3])

$$s^3(t-16)+t^3(s-16)+24st(s+t-18)-2s^2t^2 = 0. \tag{A2}$$

**Non-planar cross:** (see figure 3.b, and [3])

$$\frac{1}{3}s^3t^3u^3 - 48s^3t^3u^2 + 768s^3t^3u - 4096s^3t^3 + 4096s^3$$
$$+ 8512s^2t^2u^2 - 503808s^2t^2 - 36864\left(s^2t + st^2\right)$$
$$+ 138240\left(s^3t^2 + s^2t^3\right) - 790528stu + (\text{cyclic}) = 0, \tag{A3}$$

where $s + t + u = 4$. Curiously in the region $0 < s, t, u < 4$ this curve can be written as [46]

$$\left(\frac{s}{16}\right)^{\frac{1}{3}} + \left(\frac{t}{16}\right)^{\frac{1}{3}} + \left(\frac{u}{16}\right)^{\frac{1}{3}} = 1. \tag{A4}$$

**Green curve:**

$$s^2t^2 - 16s^2t - 32st^2 + 224st + 256(t - 1)^2 = 0. \tag{A5}$$

**Blue curve:**

$$(s - 16)^3t^2 + (s - 4)(s - 16)^3t - 16((s - 10)s + 32)^2 = 0. \tag{A6}$$

Curves (A1), (A5) and (A6) were found using the 2-particle kernel. See appendix B.

## Appendix B: Landau curves from 2-particle unitarity

As explained in section II, we can assign graphs to singularities that follow from continuation of unitarity. They represent how unitarity integrals relate singularities of sub-graphs to singularities of the bigger graph. In this appendix we demonstrate such explicit relation directly at the level of the Landau curves, without using the Landau equations. It follows from a detailed analysis of the two-particle unitarity integral – the so-called Mandelstam kernel, see [2] for details.

Consider a graph **AB** that can be split into two sub-graphs, **A** and **B**, by a 2-particle cut as illustrated in figure 7. In this case, the Landau curve of **AB** ($t_{\mathbf{AB}}(s)$)

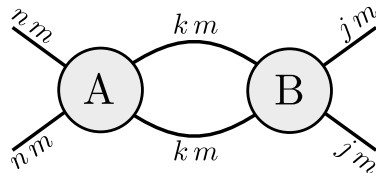

FIG. 7. The Landau curve of a graph **AB** that has a 2-particle cut is related to the singularities of sub-graphs **A** and **B**. The numbers indicate the mass of "on-shell" particles as integer multiples of $m$.

can be determined in terms of the curves of **A** ($t_{\mathbf{A}}(s)$) and **B** ($t_{\mathbf{B}}(s)$) from the following equation

$$\lambda_{n,j}(s, t_{\mathbf{AB}}(s)) = \lambda_{n,k}(s, t_{\mathbf{A}}(s)) \times \lambda_{k,j}(s, t_{\mathbf{B}}(s)), \tag{B1}$$

where

$$\lambda_{n,j}(s, t) = z_{n,j}(s, t) + \sqrt{z_{n,j}(s, t)^2 - 1}, \tag{B2}$$

and

$$z_{n,j}(s, t) = \frac{s - 2(n\,m)^2 - 2(j\,m)^2 + 2t}{\sqrt{s - 4(n\,m)^2}\sqrt{s - 4(j\,m)^2}}, \tag{B3}$$

is the cosine of the scattering angle between incoming particles of mass $n\,m$ and outgoing particles of mass $j\,m$.

To illustrate (B1) consider the leading elastic Landau curve that is plotted in gray in figure 5. It is represented by the graph on figure 8 and has a single 2-particle cut

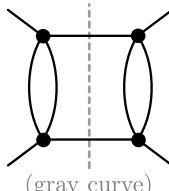 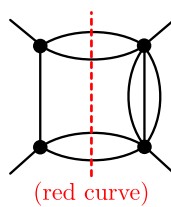

(gray curve)          (red curve)

FIG. 8. Diagrams with a single $s$-channel cut which splits each diagram into a pair of $t$-channel bubbles. The Landau curves of these diagrams follows from the singularities of the bubbles, which are simple normal thresholds. The Landau curve of the diagram on the left (gray curve in figure 5) is given by equation (B4) while the diagram on the right (red curve in figure 5) has Landau curve given by equation (B7).

along the $s$-channel. Since either sub-diagram has a normal threshold at $t = 4m^2$ and all legs have the same mass $m$, the corresponding Landau curve is given by

$$\lambda_{1,1}(s, t_{\text{gray}}(s)) = \left(\lambda_{1,1}(s, 4m^2)\right)^2, \tag{B4}$$

which in polynomial form reads

$$(s - 4m^2)(t - 16m^2) - 64m^4 = 0. \tag{B5}$$

Swapping $s \leftrightarrow t$ leads to the leading elastic curve that follows from $t$-channel unitarity. Both are represented in gray in figure 5.

Note that equation (B1) can be iterated. By gluing **AB** to a new diagram **C** one can express the Landau curve of **ABC** as

$$\lambda_{n,j}(s, t_{\mathbf{ABC}}(s)) =$$
$$\lambda_{n,k}(s, t_{\mathbf{A}}(s)) \times \lambda_{k,l}(s, t_{\mathbf{B}}(s)) \times \lambda_{l,j}(s, t_{\mathbf{C}}(s)). \quad \text{(B6)}$$

One may further iterate (B6) by gluing more sub-diagrams. If all sub-diagrams **A**, **B**, **C**, ... are taken to be $t$-channel bubbles then the full graph **ABC**$\cdots$ becomes a ladder diagram. Every Landau curve belonging to the elastic region $4m^2 < s < 16m^2$ is represented by a ladder diagram, and every elastic curve can be computed accordingly (see section 3.5 of [2]).

Interestingly, the 2-particle kernel may also be used to determine some of the Landau curves in the multi-particle regime $s, t > 16m^2$. This is because a bubble diagram with $n$ legs of mass $m$ is indistinguishable from a graph where this bubble is replaced by a single on-shell particle of mass $nm$ (see appendix D). We may therefore use equation (B1) to determine the red Landau curve in figure 5 whose diagram and cut is represented in figure 8

$$\lambda_{1,1}(s, t_{\text{red}}(s)) = \lambda_{1,2}(s, m^2)\,\lambda_{2,1}(s, 9m^2). \quad \text{(B7)}$$

Similarly, we may use equation (B6) to compute the blue and green Landau curves in figure 5, according to the slicing in figure 9.

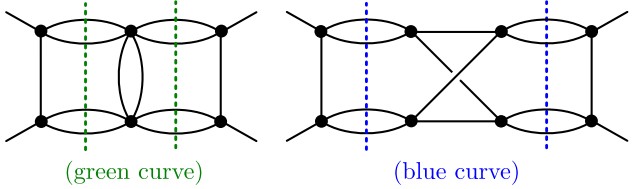

(green curve)          (blue curve)

FIG. 9. Slicing of multi-particle graphs with 2-particle cuts where each cut bubble corresponds to a particle of mass $2m$. equation (B6) may be used to find the Landau curves represented by these diagrams. Green is given by equation (B8) and blue follows from eqs. (B9) and (B10).

The green Landau curve reads

$$\lambda(s, t_{\text{green}}(s)) = \lambda_{1,2}(s, m^2)\lambda_{2,2}(s, 4m^2)\lambda_{2,1}(s, m^2), \quad \text{(B8)}$$

which, in polynomial form, is given by equation (A5).

The blue Landau curve can be expressed in terms of the Landau curve $t_*(s)$ of the sub-diagram in the middle (see figure 10),

$$\lambda(s, t_{\text{blue}}(s)) = \lambda_{1,2}(s, m^2)\lambda_{2,2}(s, t_*(s))\lambda_{2,1}(s, m^2). \quad \text{(B9)}$$

We can now relate $t_*(s)$ to the Landau curve of a simple box diagram by crossing $s$- and $u$-channels.

Using (B1) we can find the Landau curve of the box diagram and $t_*(s)$ is the solution of

$$\lambda_{2,2}(u, t_*(s)) = \lambda_{2,1}(u, m^2)\lambda_{1,2}(u, m^2), \quad \text{(B10)}$$

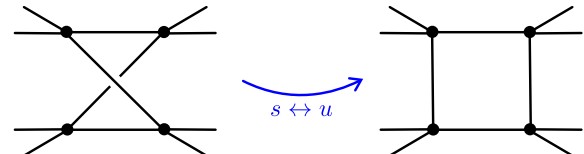

FIG. 10. Crossing $s \leftrightarrow u$ channels relates the Landau curve of the middle sub-graph of (Blue) in figure 9 with the curve of a simple box diagram. To each external vertex a bubble is connected, which is equivalent to considering $2 \to 2$ scattering of particles with mass $2m$.

where in the formula above we set $u = 16m^2 - s - t_*(s)$. Plugging $t_*(s)$ back into (B9) leads to the blue curve in figure 5, which is expressed in polynomial form in (A6).

For the graphs in this section, we have explicitly checked that the obtained Landau curves correspond to the $\alpha$-positive solutions of the Landau equations. However, this is not always the case. For example, the graph in figure 11 does not admit an $\alpha$-positive solution. Applying to it the procedure described in this section produces a Landau curve which we believe is on the second sheet.

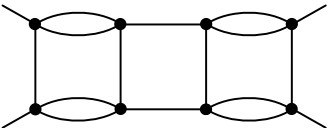

FIG. 11. A graph whose Landau curve (more precisely, its leading singularity) can be found using the method described in this section. The Landau curve sits between the blue and red curves in figure 5. Given that it not $\alpha$-positive, we do not expect it to be on the physical sheet. In the graph selection procedure, this graph was discarded by identification of a trivial box (see figure 19).

## Appendix C: Landau equations and automorphisms

Here we briefly review the standard derivation of the Landau equations for the Feynman diagrams, see for example [9] and [10] for a recent review. A generic Feynman integral with trivial numerators takes the form

$$F = \int \prod_{j=1}^{L} d^d k_j \int_0^1 \prod_{i=1}^{P} d\alpha_i \frac{\delta(1 - \sum_i \alpha_i)}{\psi^P}, \quad \text{(C1)}$$

where $L$ are the number of loops, $P$ the number of internal lines and the denominator reads

$$\psi = \sum_{j=1}^{P} \alpha_j (k_j^2 - m_j^2), \quad \text{(C2)}$$

where the $k_{j>L}$ momenta depend linearly on the loop momenta $k_{j\leq L}$, due to momentum conservation at each vertex.

The integration over the loop momentum can then be readily done and yields

$$F = \int\limits_0^1 \prod_{j=1}^P d\alpha_j \, \frac{\delta(1 - \sum_{j=1}^P \alpha_j) \, C^{P-2L-2}}{D^{P-2L}} \,, \qquad \text{(C3)}$$

with

$$C = \det a_{ij} \,, \qquad D = \det \begin{pmatrix} a_{ij} & -b_j \\ -b_j & c \end{pmatrix} \,, \qquad \text{(C4)}$$

where $i, j = 1, \dots, L$ and

$$a_{ij} = \frac{1}{2} \frac{\partial^2 \psi}{\partial k_i \partial k_j} \bigg|_{k=0} \,, \quad b_j = \frac{1}{2} \frac{\partial \psi}{\partial k_j} \bigg|_{k=0} \,, \quad c = \psi|_{k=0} \,. \tag{C5}$$

As the integral is analytically continued in the Mandelstam variables, the contour of integration may be smoothly deformed to avoid the singularities. The integral becomes singular when the contour is pinched by singularities of the integrand.

The so-called *leading singularities* occur whenever two (or more) zeros of the denominator coincide.[17]

These can be found by solving

$$\frac{\partial \psi}{\partial \alpha_i} = 0 \,, \qquad \frac{\partial \psi}{\partial k_j} = 0 \,. \tag{C6}$$

The first condition puts all internal legs on-shell, $k_i^2 = m_i^2$, while the third condition relates momenta belonging to the same loop, $l$, as

$$\sum_{i \in l} \alpha_i k_i = 0 \,. \tag{C7}$$

An equivalent form of the Landau equations is obtained for representation (C3),

$$D = 0 \,, \qquad \frac{\partial D}{\partial \alpha_i} = 0 \,. \tag{C8}$$

Note that since $D \propto \alpha_i \frac{\partial D}{\partial \alpha_i}$ is homogeneous, $D = 0$ is automatically satisfied.

There are $P + 2$ variables, $s, t$ and the $\alpha$ parameters, and $P + 1$ Landau equations, which are the $P$ pinch conditions (C8) supplemented by the normalization

$$\sum_{i=1}^P \alpha_i = 1 \,. \tag{C9}$$

These equations may be solved for $\alpha_i(s)$ and $t(s)$, the Landau curve.

In this work we made use of the form (C6) to discard trivial subgraphs (see appendix D), while (C8) is used for numerical computation of the Landau curves (see appendix E) since $D$ is an explicit function of $s$ and $t$.

As discussed in section II, we restricted ourselves to the $\alpha$-positive solutions because these occur on the undistorted contour of integration of (C1) and (C3), and are therefore likely to be on the physical sheet.[7]

We observe that non-trivial graphs (see section III) have a unique $\alpha$-positive solution, corresponding to the Landau curve represented by that graph. We assume that it is always the case.

Under this assumption, one can derive an important result which allows for dramatic simplification of the Landau equations in the search for $\alpha$-positive solutions.

Symmetries of a graph translate into symmetries of the corresponding Landau equations. Concretely, if a transformation mapping edge $i \to i'$ is an *automorphism* [8] (which also leaves $(s, t)$ invariant) then the change $\alpha_i \to \alpha_{i'}$ is a symmetry of the Landau equations (C8). Therefore, if $\alpha_i$ is a solution, then $\alpha_{i'}$ is a solution to (C8) as well.

Now, under the assumption that the $\alpha$-positive solution is unique we see that the automorphism $i \to i'$ has to map the $\alpha$-positive solution to itself. Hence,

$$\alpha_i = \alpha_{i'} \,. \tag{C10}$$

This property can be used to reduce the system of Landau equations, if one identifes the automorphisms that leave the Mandelstam invariants unchanged. Note that $(s, t)$ are left invariant if the external legs are swapped in pairs or, trivially, if they remain still.[18] If there are also automorphisms that map $s \leftrightarrow t$, the Landau curve is crossing-symmetric and further reduction is possible at the point $s = t$.

Given a graph, the exact expression for $D$ and the graph automorphisms can be found automatically using graph-theoretic tools. See appendix E for the precise implementation.

### Appendix D: Trivial subgraphs

As discussed in the main text, a trivial sub-graph is a sub-graph that either do not have an $\alpha$-positive solution or can be contracted without affecting the solution to the Landau equations. In this appendix we identify a few families of trivial sub-graphs that are composed from bubbles, triangles and boxes.

---

[17]There are also end-point singularities, corresponding to pinches at end-points of the integration contour. However, in the context of Feynman integrals, these are also leading singularities of contractions of the original graph [9].

[18]The latter case includes permutations between legs belonging to the same bubble. This implies that the corresponding parameters $\alpha_i$ have the same value, as derived in appendix D.

## Bubbles

A bubble sub-graph is a set of $n > 1$ legs connecting the same two vertices, see figure 12. When solving the associated Landau equations, we attach to them parameters $\alpha_i$ and momenta $k_i$, that are all taken to point towards the same vertex.

Each pair of these legs, $(i, j)$, form a loop and the associated Landau equation reads

$$\alpha_i k_i = \alpha_j k_j \,. \tag{D1}$$

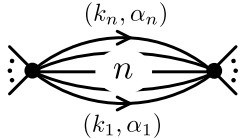

FIG. 12. A bubble sub-graph with $n$ internal lines. Each line is associated with an on-shell momentum $k_i^2 = m^2$ and a Feynman parameter $\alpha_i$. The Landau equation forces them all to be the same, making the buble equivalent to a single line of mass $n \times m$.

Using the on-shell condition $k_i^2 = m^2$ and requiring the $\alpha$'s to be positive leads to $\alpha_i = \alpha_j$. Plugging this back into (D1) leads to $k_i = k_j$.

Since all momenta are equal, the bubble diagram is indistinguishable from a single leg with mass $n \times m$.

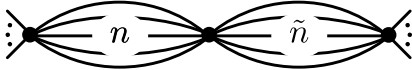

FIG. 13. A sub-graph made of a chain of two bubbles with $n$ and $\tilde{n}$ lines correspondingly. The Landau equation forces $n = \tilde{n}$, making the chain equivalent to a single bubble and hence, a trivial sub-graph.

Consider now a chain of two bubbles, one with $n$ lines and the other with $\tilde{n}$ lines, connected through a single vertex, see figure 13. Using momentum conservation at that vertex we conclude that in order to have a solution, $\tilde{n}$ must be equal to $n$. In this case however, having two bubbles instead of one impose no further constraints on the external momenta. Hence, the Landau equations for the pair of bubbles are equivalent to the ones of a single bubble.

We conclude that a bubble is trivial if to one of its vertices another bubble is connected.

## Triangles

A triangle sub-graph consists of three vertices conected by three lines, (see figure 14). To each line $i = 1, 2, 3$ we associate an independent mass $m_i$, Feynman parameter $\alpha_i$, and momentum $k_i$ that is oriented anti-clockwise

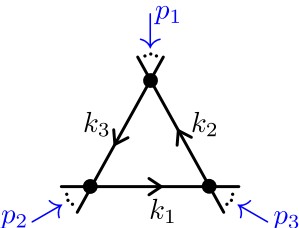

FIG. 14. A trangle sub-graph with three independent masses $m_i^2$. For $p_1^2 = (m_2 \pm m_3)^2$ the triangle is trivial.

around the loop. To each vertex we associate an incoming momentum $p_i$, with $p_3 = k_1 - k_2$, etc. The associated Landau equation reads

$$\alpha_1 \, k_1 + \alpha_2 \, k_2 + \alpha_3 \, k_3 = 0 \,. \tag{D2}$$

By dotting this equation with $k_{i=1,2,3}$, we may express it as a condition on the Gram matrix

$$\begin{pmatrix} m_1^2 & k_1 \cdot k_2 & k_1 \cdot k_3 \\ k_1 \cdot k_2 & m_2^2 & k_2 \cdot k_3 \\ k_1 \cdot k_3 & k_2 \cdot k_3 & m_3^2 \end{pmatrix} \begin{pmatrix} \alpha_1 \\ \alpha_2 \\ \alpha_3 \end{pmatrix} = 0 \,. \tag{D3}$$

Using momentum conservation, we express the Lorentz invariants in terms of the external momenta as

$$k_1 \cdot k_2 = \frac{m_1^2 + m_2^2 - p_3^2}{2} \,, \tag{D4}$$

and similarly for $k_1 \cdot k_3$ and $k_2 \cdot k_3$.

The solutions to the Landau equations (D3) can be classified into three types.

a) If one of the external momenta, say $p_1$, satisfies

$$p_1^2 = (m_2 \pm m_3)^2 \,, \tag{D5}$$

but $p_2$ and $p_3$ do not satisfy an analogous relation then the only possible solution is with $\alpha_1 = 0$. This is not an $\alpha$-positive solution.

b) Suppose two of the external momenta satisfy conditions equivalent to (D5). To have a solution with $\alpha_i \neq 0$ also the third momenta has to obey a condition equivalent to (D5) such that the product of signs in (D5) is equal to $-1$. In that case, there is a line of solutions given by the relation

$$\pm \alpha_1 m_1 \pm \alpha_2 m_2 \pm \alpha_3 m_3 = 0 \,, \tag{D6}$$

where the signs in (D6) are dictated by the corresponding signs in (D5). Requiring that $\alpha_i > 0$ selects a line of solutions with one minus and two pluses in (D6).

c) Finally, there are other solutions which are not of type (a), or (b) above, in which the triangle may be non-trivial.

While in principle a sub-triangle of type (b) may be non-trivial, we observed that until $V = 8$ for quartic graphs and $V = 5$ for sextic graphs, all such cases do not lead to a new curve. Two examples of this are given in figures 15 and 16. We assumed that this is general. Namely, that any type (b) sub-triangle belonging to a graph in the $(4_s, 4_t)$ family does not lead to a new Landau curve. Under this assumption, if at least one of the external momenta satisfies (D5) then the triangle is trivial.

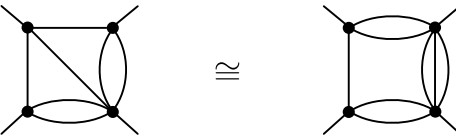

FIG. 15. An example of a sub-triangle (left) that is equivalent to a 3-particle bubble (right). The top right vertex of the triangle obeys condition (D5) with a $(-)$ sign while the top left vertex obeys this condition with a $(+)$ sign. Therefore, in order for an $\alpha$-positive solution to be possible, the momenta $p$ entering the bottom vertex must satisfy (D5) with a $(+)$ sign, i.e. $p^2 = (m + 2m)^2 = 9m^2$, which makes the triangle equivalent to the bubble with three legs on the right.

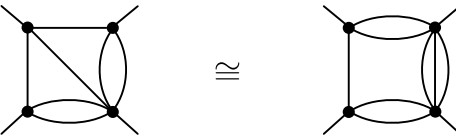

FIG. 16. The trivial "acnode" graph (left) and the accumulation graph (right) have the same Landau curve. Note that the acnode graph has two trivial triangles (top right and bottom left vertices satisfy (D5), or equivalently (D7)). No obvious graph-theoretic operation relates the graphs. It is also not clear to us how the Landau equations of the two are related, except by looking at the $\alpha$-positive solutions.

Let us now translate this condition into a graph-theoretic criterion. A generic triangle will have bubbles as internal edges, (see figure 17). As shown previously, a bubble is equivalent to single leg of mass $n_i m$, where $n_i$ is the number of bubble legs.

Suppose that we further attach an external bubble to one of the vertices of the triangle, say the vertex where $p_3$ in figure 14 enters. Condition (D5) for a triangle to be trivial then reads

$$N_3 = |n_1 \pm n_2|, \qquad (D7)$$

where $N_3$ is the number of legs in the external bubble and $n_1$ and $n_2$ are the number of legs in internal bubbles that are attached to the vertex.

## Boxes

A box sub-graph consists of four vertices, such that each vertex is connected to two other vertices, see figure

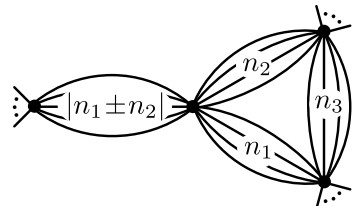

FIG. 17. A triangle is trivial if two of its internal bubbles, with $n_1$ and $n_2$ legs, are connected to an external bubble with $|n_1 \pm n_2|$ legs.

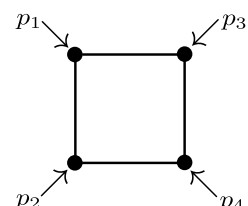

FIG. 18. Box graph. Besides the "masses" $p_i^2$ there are also Mandelstam invariants $s = (p_1 + p_2)^2$ and $t = (p_1 + p_3)^2$ that participate in the Landau equations. This makes it hard to find a generic graph-theoretic condition for the box graph to be trivial.

18. For the box graph there are 4 external momenta $p_i$ that flow into each vertex. Now, the kinematical invariants are not only the 'masses' $p_i^2$ but also the Mandelstam invariants $s = -(p_1 + p_2)^2$ and $t = -(p_1 + p_3)^2$.

Because the Landau equations now involve $s$ and $t$, a generic graph-theoretic condition for the box graph to be trivial is harder to find. However, for our purposes we do not need to discard trivial sub-boxes to quench the growth of graphs (identification of trivial triangles and bubbles allied with requiring all legs to be cut by 4-particle cuts is enough, see table I.). Rather, after step 4 of the graph selection procedure described in section III there are still trivial graphs remaining. We were able to roughly discard half of them by identifying a trivial sub-box (see table I). The remainder was eliminated by numerical search for an $\alpha$-positive solution (see appendix E).

In figure 19 we present the boxes that were found to be trivial by explicitly solving the Landau equations (see [9]). Let us emphasize once again, that we call them trivial because the corresponding Landau equations do not admit an $\alpha$-positive solution. It does not mean that these boxes do not have a nontrivial double discontinuity.

## Appendix E: Graph-theoretic implementation

In this appendix we describe how the graph selection procedure outlined in section III is implemented in detail.

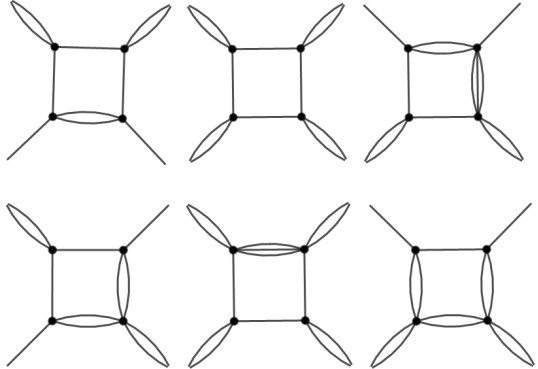

FIG. 19. A few examples of trivial boxes. We have explicitly checked that the presented graphs do not admit an $\alpha$-positive solution to the Landau equations.

### Graph generation (step 1)

To obtain all graphs with $V$ vertices and a certain maximal vertex degree, we start by generating all vacuum bubbles with $V + 1$ vertices and the same maximal degree. We then remove a quartic vertex. The four legs that were connected to it become the external legs of the scattering graph (see figure 20).

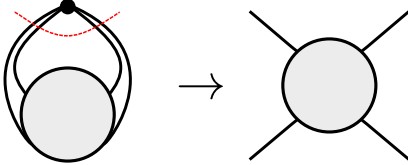

FIG. 20. Excision of a quartic vertex from a vacuum bubble (left) leads to a graph with 4 external legs (right).

To generate vacuum bubbles we make use of *nauty and Traces* [13], in particular the *geng* and *multig* commands. The *nauty/geng* command generates all simple non-isomorphic graphs with a given number of vertices $V$ and minimum and maximum vertex degrees $d_s$ and $D_s$.[19] The *nauty/multig* command takes a simple graph and turns each edge into a bubble (or multi-edge), according to maximum vertex degrees $D_m$ and maximum edge multiplicity $M$. It outputs all possible graphs with bubbles out of that simple graph. See the documentation [13] for more information.

Finally, the output of *nauty/multig* (the *adjacency matrix* [8] of each graph) is inserted into `mathematica`. Using the package *igraph/M* [14, 15] and default tools we remove a quartic vertex from the vacuum bubble to obtain a graph with 4 external legs. This procedure is exemplified for the accumulation graph in figure 21.

———————

[19] A simple graph is a graph without bubbles, only single edges [8], (see figure 21).

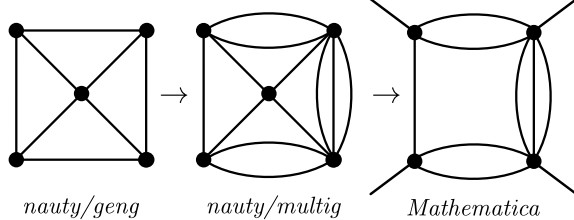

$$nauty/geng \qquad nauty/multig \qquad Mathematica$$

FIG. 21. Simple vacuum bubbles are generated using *nauty/geng* (left). Single lines are then replaced by bubbles in all possible ways using *nauty/multig* (center). Removal of quartic vertices (using *Mathematica*) leads to graphs with 4 external legs (right).

We now describe how the physical graph selection criteria described in III constrains the parameters $D_s$, $d$ and $D_m$ and $M$ that enter into *geng* and *multig*, respectively.

- $D_s = D_m = D$. It is clear that if we are looking for graphs with 4 external legs with maximum vertex degree $D$ then we can choose $D_s = D_m = D$ as long as $D \geq 4$ (so that there is a quartic vertex that can be removed from the vacuum bubble to generate the graph with 4 external legs). This condition is guaranteed because we are interested in $D = 4, 6, 8, ....$

- $d_s = 3$. Since we are only interested in vacuum bubbles it is clear that $d_s > 1$. Taking $d_s = 2$ will generate quadratic simple vertices which, when run through *multig*, will give rise to bubble chains (see figure 22) which are trivial sub-graphs (see appendix D). Thus, we should take $d_s = 3$. Indeed, in figure 21 we see that the accumulation graph comes from a simple graph with cubic and quartic vertices (left).

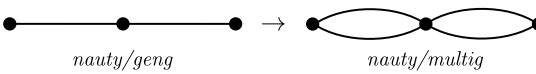

$$nauty/geng \qquad nauty/multig$$

FIG. 22. Simple quadratic vertices (left) give rise to bubble chains (right).

- $M = 3$. $M$ is the maximum number of internal lines that a bubble can have. Since we are only interested in graphs with 4-particle cuts it is clear that $M \leq 4$ suffices. Taking $M = 4$ will generate 4-legged bubbles, which can be cut by a 4-particle cut, however for such graphs there will be no cut on any other channel (see figure 23).

We import the vacuum bubbles generated by *multig* into `mathematica` and keep the ones with even degree vertices.

Finally, we discard graphs which only have a cut in one of the channels. This can be done at this stage without explicitly performing the cuts in the following way.

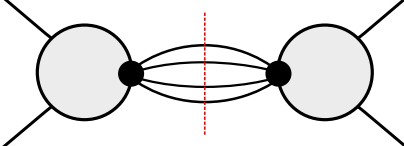

FIG. 23. A graph with a 4-legged bubble has a 4-particle cut on one channel. However, there is no space left for a cut on the other channel.

First, we require all graphs to be *bi-connected* [8]. A bi-connected graph can only be disconnected by removing two vertices. A non-bi-connected graph can be disconnected by removing a single vertex. Contracting the bubble in figure 23 leads to a generic non-bi-connected graph with arbitrary 'gray blobs' connected by a single vertex. Similarly to the original graph, it only has a cut in one of the channels.

Second, we require that the quartic vertex that is excised from the vacuum bubble does not have any bubble incident to it. This avoids the scenario represented in figure 24 where two external legs, which were originally part of a bubble, become incident to the same vertex. Such graphs also only have a cut in one of the channels. In practice, we can avoid generating such graphs by only excising vertices which were also quartic vertices in the original simple vacuum bubble.[20]

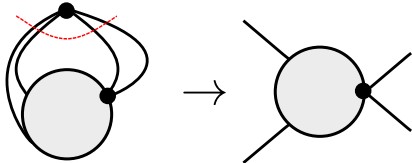

FIG. 24. Excising quartic vertices which are incident to bubbles (left) leads to graphs with cuts on only one single channel (right). Note that a quartic vertex which is incident to a bubble is a cubic simple vertex.

### Graph selection (steps 2-5)

Here we describe how steps 2-5 of the graph selection procedure described in section III are implemented in detail. We make use of `mathematica` and the package *igraph/M* [14, 15].

--------

[20]For example, in figure 21 we have 3 quartic vertices on the vacuum bubble in the center. However, only the central vertex leads to a graph with cuts on both channels. It is also the only quartic vertex in the simple vacuum bubble (left).

#### a. Trvial subgraphs

The function *IGTriangles* finds all triangles contained within any graph. Once the triangles are identified we select the cubic simple vertices (the vertices which can have an external bubble attached to it). Then, the *incidence list* [8] lists all incident edges to a given vertex. From this we can explicitly check condition (D7). If any of the vertices satisfies (D7) we have identified a trivial triangle and the graph is discarded.

A similar approach is taken to find boxes, except that there is no dedicated command to find boxes. We make use of *FindCycle* to find 4-cycles (i.e. boxes). We then compare the boxes with any of the trivial boxes in figure 19 using *IGIsomorphic*. In fact, given the handful amount of graphs after step 4 (see table I) one can just identify the trivial boxes by visual inspection.

#### b. Cuts

We are interested in minimal[6] cuts that separate external legs in pairs. There are three possible arrangements between pairs of external legs, corresponding to $s$-, $t$- and $u$-channels. To explicitly find these cuts for a given graph we apply the following procedure (depicted in figure 25).

1. Identify the 4 vertices to which the external legs are incident. We call these *external* vertices.

2. Connect the external vertices in pairs, in all three possible ways ($s$, $t$ and $u$-channels).

3. Find the source-to-sink [8, 47] minimal cuts which separate a connected pair of external vertices (source) from the other pair of external vertices (sink). Here we use *IGMinimumCutValue* or *IGFindMinimalCuts* (see below).

4. The cuts of the original graph can then be obtained by matching the cut legs found in the previous step.

In step 3 of the graph selection procedure we only ask if there is a 2 or 4-particle cut in at least two channels. For this we make use of the fast *IGMinimumCutValue* which gives the size of the smallest cut.

In step 4 we ask if all legs can be cut by a 2 or 4-particle cut. Here, we make use of *IGFindMinimalCuts* and select the cuts of size up to 4. If every internal leg is contained in (the union of) these cuts we select that graph.

### Numerical search for $\alpha$-positive solutions (step 6)

Here, we describe how the final step of the graph selection procedure is implemented in practice. At this stage no more than a handful number of graphs exists (see table I). Explicit numerical search for a solution to the Landau equations is therefore feasible.

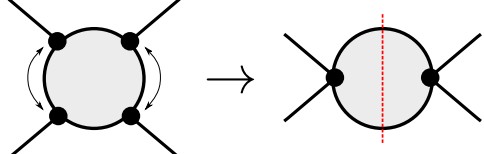

FIG. 25. A cut can be found by connecting external vertices (represented in black) in pairs. Each pair turns into two vertices, *source* and *sink*, and the cut separating external legs becomes a *source-to-sink cut* [8] which can be found using graph-theoretic algorithms [47].

We implement this step in `mathematica` (with the package *igraph/M* [14, 15]). We are given as input a graph, and the output is an $\alpha$-positive solution (if found) of the Landau equations at some fixed value of $s$. We in particular searched along the $s = t$ line point due to the existence of enhanced symmetry and consequent reduction of the Landau equations for some graphs (see appendix C). We also set $m = 1$ without loss of generality.

The algorithm is as follows.

1. Each bubble of $n$ internal legs is replaced by a single line with mass $n$. The graph becomes a simple graph.

2. A random direction is assigned to the internal edges of the graph, the corresponding *incidence matrix* [8] is found and momentum conservation at each vertex follows (see below for more details).

3. The momentum conservation equations are solved in terms of a set of independent momenta (the loop momenta). Then, the on-shell action $\psi$ as defined in eq. (C2) is computed and from there the discriminant $D$ is found from eqs. (C4) and (C5).

4. The Landau equations follow from (C8). Since we are searching for the $\alpha$-positive solution under the assumption that it is unique. We can relate different $\alpha_i = \alpha_{i'}$ if the graph is automorphic under the map $i \to i'$ (see appendix C). The automorphisms are found using *GraphAutomorphismGroup*.

5. To solve the Landau equations numerically, we square the LHS of (C8) and sum over $i$ (after the reduction described in the previous step is performed). The solutions to the Landau equations will be the minima of $\sum_i \left( \frac{\partial D}{\partial \alpha_i} \right)^2 = 0$. We perform a random search using *FindMinimum* with random starting points $\alpha_i \in (0, 1)$ and $s \in (-1000, 1000)$. The search stops when an $\alpha$-positive solution is found. A maximum of 1000 attempts was set.

Naturally, as the number of vertices increases, the system of equations becomes bigger, and the search for solutions becomes slower. Fortunately, at step 6 of the graph selection (see table I), the majority of graphs for which

this procedure was implemented enjoy some degree of symmetry, which drastically reduces the computing time. For example, $\mathbb{Z}_2$ symmetry roughly halves the number of independent $\alpha_i$'s in the Landau equations of a generic large graph.

For $V > 8$ all quartic graphs after elimination of trivial boxes (step 5) consist of triangle chains depicted in figure 4 and slight variations. For one particular variation (see figure 26) there is no automorphism relating different $\alpha_i$. For graphs with $V \geq 8$ belonging to this family we were not able to perform 1000 attempts . However, we believe that any graph belonging to this family is trivial given that for $V = 4, 5, 6$ this can be proven analytically (identification of trivial triangle or box) and for $V = 7$ no solution was found in 1000 attempts.

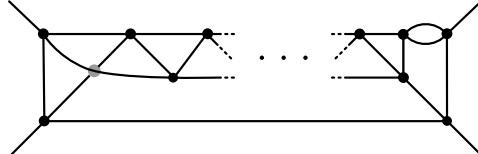

FIG. 26. Asymmetric variation of the triangle chain represented in figure 4. It is obtained by expanding into a bubble one of the vertices connecting to an external leg. This variation exists for both planar and non-planar iterations (represented by the gray vertex). For the first three iterations one can prove analytically that such graphs are trivial. For the fourth iteration, no solution was found in 1000 attempts. For the succeeding iterations we do not have a numerical argument for the absence of $\alpha$-positive solution.

*Incidence matrix and momentum conservation*

The incidence matrix $a_{ij}$ is defined as $a_{ij} = \pm 1$ if edge $j$ is incident and directed into $(+)$ or out of $(-)$ vertex $i$, and $a_{ij} = 0$ if edge $j$ and vertex $i$ are not incident (see figure 27).

It is instructive to consider a particular example. Consider the generic box graph in figure 28. Its incidence matrix is written in table III.

A few comments are in order. Note that for internal legs, the entries in the corresponding column add up to 0, while for an external legs we get $+1$ or $-1$ if the leg is incoming or outgoing, respectively.

The degree of a vertex is given by summing over the absolute value of the entry of the corresponding row. For the box graph we confirm that all vertices are cubic.

$$a_{ij} = \begin{cases} +1 & \text{if } i \bullet \!\!\!\longleftarrow\!\!\! j \\ -1 & \text{if } i \bullet\!\!\!\longrightarrow\!\!\! j \\ 0 & \text{if } i \bullet \!\!\!\longrightarrow\!\!\! j \end{cases}$$

FIG. 27. A graph can be represented in terms of the incidence matrix defined above.

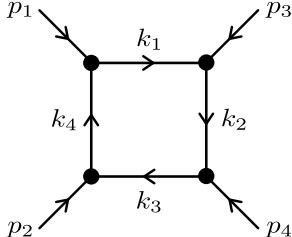

FIG. 28. Directed box graph.

| $a_{ij}$ | $p_1$ | $p_2$ | $p_3$ | $p_4$ | $k_1$ | $k_2$ | $k_3$ | $k_4$ |
|---|---|---|---|---|---|---|---|---|
| 1 | +1 | 0 | 0 | 0 | −1 | 0 | 0 | +1 |
| 2 | 0 | +1 | 0 | 0 | 0 | 0 | +1 | −1 |
| 3 | 0 | 0 | +1 | 0 | +1 | −1 | 0 | 0 |
| 4 | 0 | 0 | 0 | +1 | 0 | +1 | −1 | 0 |

TABLE III. The incidence matrix $a_{ij}$ of the box graph in figure 28. The columns are labelled according to momentum flowing on the corresponding edge, while the vertex $i$ is labelled according to the incoming external momentum $p_i$.

Importantly, the incidence matrix directly encodes momentum conservation at each vertex. If we multiply each column of the incidence matrix by the momentum that flows on the corresponding edge and sum over each line we obtain momentum conservation on that vertex.

$$\text{Momentum conservation at } i: \quad \sum_j a_{ij} q_j = 0\,, \quad \text{(E1)}$$

where $q_j$ is the momentum flowing on edge $j$.

Applying equation (E1) to the incidence matrix (given in table III) we find the expected relations

$$p_1 - k_1 + k_4 = 0\,, \qquad p_2 + k_3 - k_4 = 0\,,$$
$$p_3 + k_1 - k_2 = 0\,, \qquad p_4 + k_2 - k_3 = 0\,.$$

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
