# Peer review of "Probing multi-particle unitarity with the Landau equations"

_SciPost Physics_

## Round 1 · Referee Report · Anonymous (Referee 4) · 2022-3-9

Report
Dear Editor,
the authors study the analytic structure of the 2>2 scattering amplitude in complex s and t. While at t=0 it is known that the amplitude is analytic in s, for finite t this remains an open problem. Unitarity implies the existence of a necessary s-t double discontinuity which the authors investigate via Landau equations. Using graph-theoretical methods they identify the end-points of possible branch cuts in s-t, up to the 4-particle threshold. They identify a small class of graphs which contributes non-trivially to the discontinuity, and compute the associated Landau curves.
This work represents a small but important step in the ambitious program of solving the unitarity equations non-perturbatively. Indeed the modern incarnation of the S-Matrix Bootstrap relies on an extended s-t analyticity domain for which the present work is relevant. In addition, the well-written manuscript documents extremely well the work done and carefully positions it in relation to previous (sometimes almost forgotten) literature. Therefore I’m happy to recommend it for SciPost publication.
I have a few comments that could contribute to making the manuscript accessible to a wider audience.
1) I find that the text lacks a little bit in its introduction: the first small paragraph fails to bring enthusiasm to the non-expert reader (some of it is left for the discussion, but might perhaps already be introduced here). Perhaps reference to the Mandelstam amplitude representation could be one of the selling points — in this case perhaps attention can be drawn to the little wedge at s,t>16m^2 that still lies below the first landau curves, and the importance of assessing whether it is free of non-analyticity.
2) In section 2, graphs are used to keep track of the discontinuities of Landau equations. The same graphs are found in Feynman diagrams, when some of the legs (off-shell) are contracted; since graphs and diagrams only keep track of the kinematics, the authors focus on Feynman Diagrams. I found this part slightly confusing: perhaps the notion of graph (and the difference with Feynman diagram) could be made clearer. Also, would this hold also for composite particles like pions? (I am not sure whether this is already addressed with the requirement that there be no bound-states…perhaps it depends on how bound-states are defined: poles below threshold?).
2) When Landau equations are introduced, perhaps it might be nice to remind that momenta can be complex and that point 3 (page 3) is reminiscent of the integration over Feynman parameters.
3) Perhaps in section 3, when the vertex-degree D is introduced, it might be worth recalling that (because of the notion of contractions introduced earlier), a high D can be found also in theories where the fundamental interaction is smaller than D.
4) In the conclusion, I find the discussion on lightest-particle-maximal-analyticity extremely interesting, although perplexing. How is it at all possible that all the subgraphs give vanishing contributions in bigger graphs? Could the authors give a few more hints of how they think this takes place?
5) Could the authors comment on what would happen if masses where not taken equal, or if the theory possess bound-states?
Typos:
point 2 in page 3, the m_i^2 shall have no index i.
page 11, 4 lines above start of appendix C: Given that it IS not alpha positive…
Author: Miguel Correia on 2022-06-22 [id 2603]
(in reply to Report 1 on 2022-03-09)(Attached updated manuscript)
Attachment:
Author: Miguel Correia on 2022-06-22 [id 2601]
(in reply to Report 1 on 2022-03-09)
We thank the referee for the careful reading of the manuscript and the useful comments and suggestions.
We now respond in detail to each comment:
1) Indeed, as suggested, mention is now made in the introduction to the question of the non-perturbative support of the double discontinuity (last paragraph in page 1).
2) We agree. The point is that Feynman diagrams have non-perturbative singularities, i.e. kinematic singularities which are due to unitarity, and which should be present in the full amplitude as well. The distinction between graph and Feynman diagram is that the former is a purely kinematic object where every line corresponds to the exchange of a on-shell particle, whereas the latter is generically an off-shell perturbative object, where vertices are associated with a specific Feynman rule, and internal lines are typically associated with fundamental fields in a Lagrangian. Hopefully this distinction is now made clear in the last paragraph of left column of page 3 and fourth paragraphs on the right column of page 3. Footnotes 8 and 9 were also added.
2) "complex-valued" was added to point 1. on the right column of page 3.
3) Footnote 12 was added in this context.
4) We don't exactly know what is the underlying mechanism that protects such anomalous features from lightest particle scattering. This seems to be the case however. We have added footnote 19 with a concrete example. We find an alpha-positive solution for the equal mass case, which we would have found had we relaxed Z2 symmetry. However, as soon as internal masses are increased, we dont find any alpha-positive solution. Somehow it seems that such anomalous features are sent to the second sheet if the external states are the lightest.
5) This is a interesting disccusion which the manuscript was originally lacking. We have consequently added two paragraphs to the end of section IV C., and another paragraph to the end of section IV E. In section IV H we also have included the second bullet point. In essence, the general mass case introduces unpleasant features such as anomalous thresholds which, from a practical point of view, makes our graph selection criteria harder to implement. Moreover there may be complex singularities in the physical sheet which evade an alpha-positive analysis, which we have restricted ourselves to.
Typos were corrected.
Author: Miguel Correia on 2022-06-22 [id 2602]
(in reply to Report 2 on 2022-03-24)Reply to referee report 2:
We thank the referee for the careful reading of the manuscript and the useful comments and suggestions.
We now respond in detail to each comment: 1- Our analysis was restricted to the region where s,t are both real and larger than 16m^2. This means that the Landau curves that were found all lie on the real axis, in particular on top of the usual normal thresholds. Hopefully, this is now clear in the definition of LPMA (page 6 at the bottom).
2- Some diagrams in fig. 5 have internal bubbles, which are equivalent to heavier particles. In particular, the accumulation graph (red) exchanges an intermediate particle of mass 3m. All these curves obey monotonicity and are asymptotic to normal thresholds. This is however not the case if the external states become heavier, due to the presence of anomalous thresholds. Hopefully the two new paragraphs at the end of section IV. C make this clear (page 8).
3- Yes, we believe this to be a general feature. In fact, there is no distinction between the bubbles in Fig. 6 and single lines of heavier particles. Both yield the same Landau curves. A new paragraph was added to the end of section IV E. Also the discussion on going beyond Z2 symmetry was expanded in section IV H (1st point).
Attachment:
Landau_curves_inelastic.pdf

---

## Round 1 · Referee Report · Anonymous (Referee 3) · 2022-3-24

Strengths
1- The paper is comprehensible and well-motivated, the goals are stated clearly.
2- The organisation of the paper is optimal, with a review section, followed by the core and a more speculative section at the end. Many appendices are provided with detailed algorithms and explicit computations.
Weaknesses
1- Although the algorithms for the computations are very detailed in the appendices, it could be useful to post it on a public repository/attach it to the publication. This might help reproduce and extend the results in the future.
2- Some of the conjectures put forward in the discussion session sound a bit mysterious and need clarifications.
Report
In the paper, the Landau curves of several Feynman integrals are studied. In particular, the focus is on the integrals contributing to the four-particle cuts for which the solution of the Landau equations can be found on the physical sheet. Assuming that there are no bound states with mass smaller than $2m$ and a $\mathbb{Z}_2$ symmetry, it is argued that the set of Landau curves is complete in the kinematical region under consideration and that these curves are a non-perturbative feature of the S-matrix. Remarkably, the authors found two families of curves (corresponding to planar and non-planar graphs respectively, with a chain of triangles) that accumulate on the same Landau curve. They argue that this accumulation feature is universal above the multi-particle threshold.
Definitely, the article deserves to be published after a few clarifications.
Requested changes
1- In Section IV.A, the connection of this work with the LPMA conjecture is discussed. The statement of this conjecture, as presented in the section, is quite confusing. The definition of the physical sheet, given in the introduction, and the infinite set of Landau curves in it (besides threshold branch points and simple poles) seem to be incompatible with the statement presented in the paragraph. The authors may clarify this point.
2- In Section IV.C, a conjecture for the $\alpha$-positive Landau curves for the lightest particles scattering is proposed. This conjecture seems to be very strong beyond the four-particle cut when we consider Feynman integrals involving intermediate states with a mass larger than $2m$. The authors may discuss this further or refine their statement.
3- A related comment to the previous one is about the conjecture in Section IV.E. Do the authors expect the Landau curves associated with the graphs in Figure 6, with the internal lines being heavier states, to be also accumulation points? Also, it seems from the discussion that the conjecture goes beyond the restriction to $\mathbb{Z}_2$-symmetric scattering. The authors may refine also this statement.

---

## Editorial Decision

unknown